# Aflatoxin Biosynthesis, Genetic Regulation, Toxicity, and Control Strategies: A Review

**DOI:** 10.3390/jof7080606

**Published:** 2021-07-27

**Authors:** Rahim Khan, Farinazleen Mohamad Ghazali, Nor Ainy Mahyudin, Nik Iskandar Putra Samsudin

**Affiliations:** 1Department of Food Science, Faculty of Food Science and Technology, University Putra Malaysia, Serdang 43400, Malaysia; sirifrahim1@yahoo.com (R.K.); nikiskandar@upm.edu.my (N.I.P.S.); 2Department of Food Service and Management, Faculty of Food Science and Technology, University Putra Malaysia, Serdang 43400, Malaysia; norainy@upm.edu.my; 3Laboratory of Halal Science Research, Halal Products Research Institute, University Putra Malaysia, Serdang 43400, Malaysia; 4Laboratory of Food Safety and Food Integrity, Institute of Tropical Agriculture and Food Security, University Putra Malaysia, Serdang 43400, Malaysia

**Keywords:** aflatoxins, biosynthetic pathway, genetic regulation, toxicity, control strategies

## Abstract

Aflatoxins (AFs) are highly toxic and cancer-causing compounds, predominantly synthesized by the *Aspergillus* species. AFs biosynthesis is a lengthy process that requires as minimum as 30 genes grouped inside 75 kilobytes (kB) of gene clusters, which are regulated by specific transcription factors, including *aflR*, *aflS*, and some general transcription factors. This paper summarizes the status of research on characterizing structural and regulatory genes associated with AF production and their roles in aflatoxigenic fungi, particularly *Aspergillus flavus* and *A. parasiticus*, and enhances the current understanding of AFs that adversely affect humans and animals with a great emphasis on toxicity and preventive methods.

## 1. Introduction

Aflatoxins (AFs) are secondary metabolites predominantly synthesized by *Aspergillus flavus* and *A. parasiticus.* They are highly toxic, mutagenic, carcinogenic, immunosuppressive compounds with severe detrimental effects on the human liver [1]. AFs contamination in food products is a worldwide issue and a possible risk to human and animal health [2,3]. The threat of AFs to human and animal health was first recognized after their identification as a causal agent of turkey X infection in poultry in the UK. The toxin was detected in feeds, and its properties and biological impacts were then investigated [4]. The term AF was given to the toxin since it was produced by *A. flavus*. In tropical and subtropical regions, billions of people were impacted by AFs adversely by consuming contaminated foods and water [5]. AF exposure is closely related to increased risk of hepatocellular carcinoma (HCC), AIDS, stunting, and malnutrition in children in America, Asia, and Africa [6,7,8,9,10,11,12,13,14]. Contamination of corn, peanuts, rice, and cottonseed with AF has been linked to agricultural losses and increased liver cancer incidence in Central America, Africa, and Asia. The study on the mechanism of AF production directly influences our capacity to diminish AF adulteration in the food supply chain.

Consequently, AF production has been developed into the most extensively studied biological activity. In the early 1990s, molecular biologists began to pay attention to AF biosynthesis, and the primary genes responsible for AF-production (*nor-1 and ver-1*) were identified and transcribed [15]. Later, the complete gene cluster for AF was identified [16,17]. Thus far, several genes, proteins, and regulatory mechanisms have been extensively investigated. Thus, the AF’s biosynthetic pathway helped to develop an outline for the production of mycotoxins and metabolic pathways in eukaryotic organisms. This paper will review the characterization and functions of structural genes involved in the production of AFs, genetic regulation and toxicity of AFs, and serval novel methods developed over the last few decades to minimize humans’ vulnerability to AFs in high-risk communities.

## 2. Biosynthetic Pathway of Aflatoxins 

Following the revelation of Turkey X disease, researchers began studying AF biosynthesis by developing ultraviolet variants [18]. Different researchers recently characterized the entire 75-kb cluster on chromosome 3′s subtelomeric locus [19]. Although the gene cluster of *A. flavus* is similar to *A. parasiticus* in terms of sequencing, they are markedly different in deletion, ranging between 0.8 kb (L-strain) and 1.5 kb (S-strain). This deletion extends from the 5′ end of *aflF*, *aflU* to the whole 279 bp intergenic loci, preventing *A. flavus* from producing AFG_1_ and AFG_2_. DNA analysis revealed that strains of *A. flavus* and *A. parasiticus* exhibit approximately 96% affinity for this gene cluster [20]. Research on *A. parasiticus* identification found that 30 genes are located in this gene cluster [21]. The genes and enzymes associated with the AF biosynthesis pathway in *A. parasiticus* are presented in Figure 1. Two substrates contribute to AF biosynthesis, known as 1-Acetyl-CoA and 9-Malonyl-coA. Here, we will discuss the genes, encoding proteins, and precursors involved in AF production.

Figure 2 demonstrates each phase of the AF biosynthesis pathway. Norsolorinic acid (NOR) is the primary step of the AF biosynthesis pathway.

### 2.1. Synthesis of Norsolorinic Acid (NOR)

Three proteins, including fatty acid synthase α (*aflA*), fatty acid synthase β (*aflB*), and polyketide synthase (*aflC*), are responsible for the production of NorS. NorS plays a vital role in the synthesis of the hexanoyl primer through integrating with malonyl-CoA molecules. Afterward, the hexanoyl primer is moved to the region of β-ketoacyl synthase [24] and combined with malonyl-CoA to form norsolorinic acid anthrone (NAA). Due to its high reactivity, this metabolite is rapidly converted into NOR by NAA oxidase [25]. NOR, an essential metabolite synthesized in the AF’s biosynthetic pathway, exhibits a red–orange color in mutant strains of *aflD* (*nor-1*) of *A. parasiticus* [26].

### 2.2. NOR Conversion to Averantin (AVN)

*AflD*, a ketoreductase, reduces the NOR 1’-keto group to the AVN 1′-hydroxyl group [27]. Even though its role is defined, the mutant strain of *aflD* does not always result in AVN formation. The other processes contributing to this reduction remain unknown at this point.

### 2.3. AVN Conversion to 5′-Hydroxyaverantin (HAVN)

*AflG*, a monooxygenase of cytochrome P450, catalyzes the breakdown of the 5′-keto group of AVN to the 5′-hydroxyl group of HAVN [28]. 

### 2.4. HAVN Conversion to Averufin (AVF)

The HAVN dehydrogenase facilitates the dehydrogenation of the HAVN’s 5′-hydroxyl group to 5′-oxide group of oxoaverantin (OAVN) [29]. The deleted *aflH* mutant consistently demonstrates its ability to synthesize OAVN, suggesting the involvement of other potential mechanisms. In contrast, *aflK* is an OAVN cyclase that catalyzes the dehydration of 5′-oxide of OAVN to form the 2′-5′ AVF [30].

### 2.5. AVF Conversion to Versiconal Hemiacetal Acetate (VHA) 

Being a cytochrome P450 oxidoreductase, *aflV* can reduce the hydride group of AVF [31]. The projected compound becomes hydrated, while aflI presumably functions as an oxidoreductase [32]. On the other hand, *aflW* monooxygenase is vital for incorporating the O_2_ atoms within the 4’-5′ ketone groups of HAVN, forming VHA.

### 2.6. VHA Conversion to Versiconal (VAL)

*AflJ*, an esterase enzyme that stimulates VHA acetate eradication, results in converting the latter into VAL [33].

### 2.7. VAL Conversion to Versicolorin-B (VERB)

*AflK*, a cyclase that catalyzes the cyclodehydration of VAL into VERB [23]. This is a crucial phase in the AF’s biosynthetic pathway as the closure of the bisfuran ring occurs at this stage. Additionally, it serves as the final precursor for the biosynthetic pathways of AFB_1_-AFG_1_ and AFB_2_-AFG_2_.

### 2.8. VERB Conversion to Versicolorin A (VERA)-AFB_1_-AFG_1_ Pathway

*AflL*, a monooxygenase of cytochrome P450, is responsible for converting the tetrahydrofuran ring to a dihydrobisfuran ring [34]. 

### 2.9. VERA Conversion to Demethylsterigmatocystin (DMST) and VERB Conversion to Dihydro Demethylsterigmatocystin (DHDMST)

*AflM*, *aflN*, *aflY*, and *aflX* are putative enzymes involved in DMST formation in the biosynthetic pathway of AFB_1_-AFG_1_ [35]. The same enzymatic steps have been suggested in the biosynthetic pathway of AFB_2_-AFG_2_, but using VERB as a substrate rather than VERA, resulting in DHDMST formation. The discrepancy amid DMST-DHDMST is comparable to that of VERA-VERB, owing to the bisfuran double bond. 

### 2.10. DMST Conversion to Sterigmatocystin (ST) and DHDMST Conversion to Dihydrosterigmatocystin (DHST)

*AflO*, an O-methyltransferase, is responsible for transmitting the S-adenosylmethionine methyl group, DMST hydroxyl group, and synthesis of DHDMST to ST and DHST based on biosynthetic pathways [36].

### 2.11. ST Conversion to O-Methylsterigmatocystin (OMST) and DHST Conversion to Dihydro-O-Methylsterigmatocystin (DHOMST)

*AflP* is a second O-methyltransferase of AF biosynthesis appropriate for ST and DHST substrates [37]. Strains of *A. nidulans* preclude the synthesis of AF as they lack the *aflP* orthologue [38].

### 2.12. OMST Conversion to AFB_1_ and DHOMST Conversion to AFB_2_

*AflQ*, another monooxygenase of cytochrome P450, transforms OMST into AFB_1_ [39]. Yu [40] suggested a comprehensive metabolic pathway in which *aflQ* is replicated in C-11 hydroxylation, while *aflLa* may serve as a source of O_2_ for the keto-tautomer 11-hydroxy of OMST. These reactions might result in the formation of 370 da metabolites. On the other hand, it is assumed that *AflMa* is responsible for demethylating the A-ring and might work in conjunction with a cytochrome P450 as a final phase of the AF biosynthesis pathway.

### 2.13. Bis. OMST Conversion to AFG_1_ and DHOMST Conversion to AFG_2_

The 370-da metabolites could serve as substrates in *aflU* oxidation, which results in the synthesis of AFG_1_ and AFG_2_ [41]. Thus, NadA and *aflF* could be suitable candidates for enhancing *aflU* activity in the production of AFG_1_ and AFG_2_.

## 3. Genetic Regulation of Aflatoxin Biosynthesis

The above-mentioned phases of AF production are regulated by certain specific transcription factors such as *aflR* and *aflS* and some general transcription factors. 

### 3.1. AflR, a Specific Transcription Factor

*AflR* is the ninth gene of the AF biosynthetic cluster that encodes the Cys_6_Zn_2_ transcriptional factor required for AF production. Figure 3 represents the structure of the *aflR* transcriptional factor.

The N-terminal part of the *aflR* (C6 cluster) includes the NLD (Nuclear Localization Domain), which is required for *aflR* movement from the cytoplasm to the nucleus [42], while the linkage portion may contribute to DNA-binding affinity. The DNA sequence is 11 bp long (5′-TCGSWNNSCGR-3′), featuring the highest binding affinity for 5′-WCGSNNNSCGA-3′. These *aflR*-binding loci are typically located at the 200 bp exterior to the translation start point of the AF gene. Upstream of the *aflR* gene’s translation start point, a partial *aflR* binding site indicates autoregulation. Other binding sites of diverse DNA binding proteins in the same intergenic region show that different regulatory networks could regulate the expression of *aflR*. Price et al. [43] analyzed 40% transcriptomes of *A. parasiticus* in its wild-type and *aflR*-mutated strains that cannot generate AFs. They discovered that the *aflR* mutant lacks most of the AF genes in the cluster except for *aflF*, Ma, N, and Na.

### 3.2. AflS, a Putative Transcription Factor

*AflS* is the 10th gene in the cluster of AF biosynthesis pathways, sharing a similar intergenic region with *aflR.* Although the knockout mutants demonstrated that *aflS* is needed for AF’s synthesis, its exact role is yet to be determined. The three possible functions of *aflS* are as follows:It may operate as an *aflR* coactivator [44], although its deletion has little effect on *aflR* transcript levels.It strongly affects the early genes involved in AF production [45].*AflS* mutants inhibit the *aflC*, *aflD*, *aflM*, *and aflP’s* transcription by up to 20 times yet do not affect the expression of *aflR*. In contrast, other researchers ruled out the effects of *aflS* on *aflM* and *aflP’s* expression.It is vital for *LaeA* to target a particular gene cluster. Furthermore, it is sensitive to temperature during incubation; henceforth, the expressions of *aflS* and *aflR* were increased by 24 times at 30 °C compared to 37 °C [46].

### 3.3. General Transcription Regulators

Seven well-known general transcriptional regulators control the biosynthetic mechanism of AFs. Each pathway is crucial to our research as it explains how specific genes of AFs are expressed or inhibited. A complex network of proteins regulates the synthesis of secondary metabolites of fungi [47]. Figure 4 shows the three essential pathways that regulate AF production.

The heterotrimeric G-protein pathways (G proteins) are general transcription regulators linked to the plasmid membrane of the cell and function as transduction impulses in reaction to foreign stimuli to maintain the cell’s physiologic conditions. The G proteins have three subunits—α, β, and γ—that abandon their function once reassembled into a trimeric form (Figure 4). The instigation occurs because of GTP binding to the G subunit. Regarding AF biosynthesis, it has been demonstrated that two subunits of Gα (*GanB* and *FadA*) prevent ST/AF synthesis in the presence of GTP through the suppression of *aflR* activity [48,49]. Nevertheless, it was shown that the Gβγ subunits (SfaD and GpgA) stimulate ST synthesis, implying that the G protein subunits analyzed have distinct functions in ST biosynthesis [50].

Moreover, the response to Reactive Oxygen Species (ROS) is a second transcription regulator. Figure 4 illustrates a suggested mode of action for such reactions. *YapA* gene’s mutation increases AF production, indicating that *YapA* may act as an inhibitor of ROS buildup. It was discovered that in the presence of ROS, four DNA-binding transcriptional factors, such as *MsnA*, *AtfB*, and *AP-1/SrrA* complexes, entangle to the specified DNA to stimulate AF production by boosting AF genes [51]. Similarly, the light-sensitive complex (*VeA*, *VelB*, and *LaeA*) is a third transcription regulator (Figure 5) that exhibits a low amount of *VeA* activity in light inside the cytoplasm. Nonetheless, *VeA* expression increased in the dark and was carried into the nucleus via the importer α carrier (KapA) [52]. *Therefore*, *LaeA* should be bound to a *VeA/VelB* compound to have an inhibitory impact over *HepA*. *HepA* is a spatial adaptor that plays a vital role in chromatins’ molecular compounds [53,54]. The *HepA*’s suppression prevents the transition of heterochromatin towards euchromatin in the *aflR* region [55]. 

Additionally, the ppoABC genes contain three distinct putative fatty acid oxygenases involved in producing oxylipins by fungi (Figure 6) [56]. The *VeA*, hydroxylated linoleic (psiα), and oleic acid (psiβ) proteins are thought to be involved in the shift from sexual to asexual reproduction in fungi [57]. A dual deletion in the ppoABC gene resulted in the inhibition of ST production, but a single loss of ppoB enhanced ST accumulation (Figure 6).

Recent research indicates that diverse Ppo oxygenases may result in the aggregation of oxylipin exterior to the cell of fungi and may stimulate the activity of G-proteins. Three additional general transcription regulators are activated in response to external stimuli (Figure 7), which are briefly discussed here. *CreA* is a zinc transcription factor that responds to carbon supply by triggering metabolic activities [58,59]. Additional characterization is necessary to have an in-depth insight into the fundamental processes. Likewise, *AreA* is the zinc transcriptional factor regulating the nitrogen pathway [60]. The *aflS-aflR* intergenic region has an AreA-binding site that may induce AF production.

Likewise, *PacC* is a transcription factor that negatively regulates the ST’s biosynthetic pathway in *A. nidulans* under alkaline conditions [61]. Its inactivation is pH-dependent and may be reversed under acidic conditions. In addition to the above-mentioned regulatory mechanisms, other processes, including the production site and excretion process, may affect AF synthesis.

## 4. Aflatoxins Toxicity

AFs are the most significant food safety concern since they are widely distributed in foods and feeds and are highly toxic. AFs carcinogenicity has long been linked with the liver, where they produce transitional metabolites; however, recent epidemiological and animal trials revealed that they were carcinogenic to organs other than the liver, including the pancreas and kidneys, bones, bladder, and central nervous system [62]. Other than that, other AF-induced long-term health impacts include anemia, malnutrition illnesses, retardation in physical and mental growth, and nervous system maturation. Despite these challenges, their modes of action need further clarity [63].

### 4.1. Chronic Aflatoxicosis

The consumption of AF-contaminated foods is typically linked with HCC and bile duct hyperplasia [64]; however, other organs, including kidneys, the viscera, lung, bladder, and bone, were also found to develop cancer once exposed to AFs. AFs also cause lung [65] and skin [66] cancer mainly through inhalation and direct contact. Other complications resulting from AF consumption include immunosuppression, mutagenicity, teratogenicity, and cytotoxicity in mammals, particularly in rats and humans [67]. Furthermore, AFs are associated with nutritional disorders, including kwashiorkor and growth faltering, possibly influencing the accumulation of iron, zinc, vitamin B, protein synthesis, and other enzyme activities [68,69]. Lower doses of AFs are often detrimental to the health, productivity, and reproduction of livestock and increased vulnerability to infections. Despite the insidious property of chronic aflatoxicosis, its health impacts are more catastrophic and costlier than acute infections. Chronic aflatoxicosis with hepatitis B (HB) has been reported to increase AFB_1_ potency up to 60-fold. According to the latest IARC Global Cancer Observatory statistics, over 841,080 new liver cancer cases and 781,631 fatalities were recorded in 2018. It equates to the age-standardized frequency of 9.3 per 100,000 persons and a fatality ratio of 93%. It is the fifth most prevalent malignancy and the primary cause of cancer-related deaths. The continents of Asia and Africa consistently produce the newest incidents, with 64,779 (7.7%) and 609,596 (72%) cases, accounting for over 80% of the total cases globally. AFB_1_ alone was expected to induce 25,200–155,000 infections each year [70,71], of which almost 40% of cases were reported in Africa, whereby AF-induced liver cancers are responsible for one-third of all liver cancers [72]. China holds the world’s highest rate of liver cancer at a country level, most of which were reported in the country’s southern part, where dietary AF exposure and HB chronic diseases are prevalent [73,74,75,76,77].

#### 4.1.1. Immunotoxicity

An increased prevalence and severity of infectious diseases and extended healing time with reduced vaccine effectiveness have established that AFs impair the innate and adaptive immune system [78,79,80,81]. Some recent studies have reported that AFBO interacts with innate and immune-competent cells in the body, influencing their reproduction and generation of immune response mediators, impeding the establishment of adaptive and innate immune systems. Research conducted for observing the toxicity mechanisms in animals and humans has discovered the immunotoxicity of AFB_1_ on human cell lines in highly exposed regions of Ghana [82,83]. Alternatively, some researchers have examined the immunotoxicity of AFs rather than AFB_1_ [84,85,86,87]. Meanwhile, a general agreement exists that low to medium levels of AFB_1_ may not have detrimental effects on the immune system, although cell-mediated immunity is highly susceptible to AFs compared to hormonal immunity [88,89]. 

#### 4.1.2. Innate Immunity

The breakdown of physiological barriers, such as epidermal and gastrointestinal mucosal tissues with pathogen invasion, has been shown in vitro and in vivo studies. For example, animal skin contact with AFB_1_ has been reported inducing various lesions, including intra-epidermal vesicle production and squamous cell carcinoma [90,91,92,93]. Pigs fed with a mixture of AFB_1_ and AFB_2_ for 28 days developed irritation and cutaneous ulcers on the nose, lips, and labial commissure of the mouth. Another study has shown that AFs impair the intestinal mechanical barrier integrity by intervening with the cell cycle or disrupting epithelial cells and tight junctions, cementing them together. These results have recently been supported by research in which a broiler chicken was fed with a feed comprising 0.6 mg AFB_1_/kg for 21 days and reported various structural and functional variations in the gastrointestinal tract, such as the contraction of mitochondria and depletion of absorptive cell goblets [94,95]. Such alterations drastically change the intestine’s ability to absorb nutrients and the innate immune response that protects against the invasion of pathogens and toxins. The impacts of AFs on immune cells, including macrophages, monocytes, natural killers, and dendritic cells, have been well established.

Additionally, AFB_1_ and AFM_1_ have been found to decrease the feasibility, multiplication, and necrosis of macrophages and cytokines’ production, including TNF-a and IL-1 [96,97,98,99]. Recently, autophagy was reported to influence the innate immune system, particularly M_1_-type macrophages, which are involved in inflammation responses induced by proinflammatory cytokines. Feeding research has also shown a reduced complement behavior in livestock and poultry at varying levels [100].

#### 4.1.3. Adaptive Immunity

The inhibition of adaptive immunity following AF’s exposure is well documented, demonstrating improved susceptibility of exposed hosts to contagious agents and weakened vaccine defense [101,102]. The epidemiological research demonstrated that vaccination failed to protect poultry from bronchitis [103] and Newcastle disease once exposed to AFs. The same types of suppressive impacts have been observed in swine, in which vaccine treatment did not defend them against *E. rhusiopathiae* once exposed to AFB_1_ [104]. It is also reported that reduced replication, activation, and lymphocyte activity are critical elements of humans’ adaptive immunity. Dose and time-dependent apoptotic impacts have been seen in human blood cells after being incubated with 3.12–2000 g/L of AFB_1_ solution for 2–72 h [105,106]. Recent research also found that AFB_1_ and AFM_1_ substantially improved the IL-8 activity, which is connected with innate immunity.

Similarly, in humans, a high level of AFB_1_ is closely associated with reduced lymphocyte percentages and plays an essential role in immunization and inflammatory responses to microbial infections. In addition, previous studies’ results indicate that AFB_1_ inhibits cell-mediated immunity in human beings, weakening their tolerance to infection [107]. It is noteworthy, however, that humoral immunity and cell-mediated immunity might not always be distinguishable. For instance, the deregulation of dendritic cell proliferation and expression of TLRs can affect both innate and adaptive immunity since such antigen-producing cells serve as critical intermediaries for both forms of the immune response [108,109].

#### 4.1.4. Teratogenicity

AF exposure to pregnant females or animals may inhibit the growth and development of embryos in the womb, leading to different health problems and pathological outcomes [110]. In Asian and African countries, mothers are highly exposed to AFs; they transmit AFs to their fetuses through blood circulation. AFs and their resultant biomarkers (AF metabolites, AF-DNA, and AF-albumin adducts) were found in both fetal cord and mother’s blood samples [111,112]. Hence, it is inferred that AFs and their metabolites are passed to the fetus metabolized by the same pathway as adults [113]. As a result, maternal risk factors greatly influence fetal growth, causing weight loss and premature delivery. An inverse correlation between birth weight and the number of suitable biomarkers in umbilical cord blood samples has been established in humans and animals [114,115,116].

On the other hand, very few studies have correlated AF consumption in pregnant ladies with early delivery and miscarriages [117]. Apart from the above stated detrimental health impacts, AF contaminated meals during pregnancy impair pregnant women’s well-being and expose their fetuses to indirect risks of congenital anomalies, including impeding placental growth, stillbirth, miscarriage, and premature birth. Additionally, AFs interfere with the availability of iron, selenium, and vitamins and result in anemia and low fetal development, or premature childbirth. However, data are scarce on the relationship between AF exposure and inflammation-related anemia among pregnant females. The data on dose, procedures and sensitivity to AFs exposure in pregnant women need further studies to improve pregnancy and delivery safety.

#### 4.1.5. Malnutrition

Along with the essential toxicological impacts discussed above, AFs cause various other detrimental health effects by overlapping processes and risk factors, including malnutrition disorders (stunting), delayed physical and mental growth, fertility problems, and nervous system disorders [118,119]. Malnutrition has garnered enormous attention because of its detrimental effects on children worldwide, especially in underdeveloped nations wherein kids suffer food scarcity. To be specific, one must ensure that kids obtain physiological and cognitive maturity and are ready for adulthood as responsible and productive persons. AF exposure deprives children of these vital micronutrients and often enhances their vulnerability to AFs, which they usually detoxify with the help of endogenous antioxidants [120,121,122]. Consequently, exposed children can experience development defects beginning from the gestational phase, resulting in stunted growth and delayed physiological and psychological development. The stunted growth in kids under the age of five in African nations has been linked to chronic exposure of AFs since they depend upon indigenous agriculture items such as corn, peanuts, and derivatives as staple foods [123]. Protein-energy malnutrition illnesses, including kwashiorkor and marasmic kwashiorkor, have also been linked to the higher level of AF exposure in various African nations [124,125,126,127]. A study on malnourished Sudanese children with kwashiorkor and marasmic kwashiorkor reported that their serum and urine samples had a substantially greater concentration of AFB_1_ than children undernourished with marasmus. The researchers concluded that kwashiorkor was related to the long-term exposure of AFs, owing to the liver injury or an etiological element of such sickness that has not yet been identified.

#### 4.1.6. Neurodegenerative Diseases

Apart from the well-established detrimental health impacts of AFs, there is a growing realization indicating that long-term AF exposure may often lead to neurodegenerative diseases. The AFBO and ROS synthesized by the CY450 enzyme and AF-induced oxidative stress interact with active molecules in neurons, inhibiting lipid and protein production and damaging fatty and polypeptide molecules. Additionally, AFs have been found to impair the mitochondrial activity of neurons, which results in apoptosis [128]. Furthermore, the discovery of AFs in kwashiorkor-deceased children’s brain tissues and their relation to cerebellar edema suggests that AFs can cross the brain–blood barriers, and penetrate the neurological system. The epidemiological research on the neurotoxicity of AFs has found AFs in human and animal nervous systems. In addition to oxidative stress, AFs encourage neurodegenerative issues by degenerating immunocompetent cells’ immune responses and producing proinflammatory situations in the brain and spinal cord.

### 4.2. Acute Toxicity

Acute toxicity is predominantly linked to AFs protein adducts since they inhibit enzymes involved in metabolic pathways, protein production, DNA replication, and immune responses. Moreover, there is a mounting indication that AFs phospholipid adducts are the primary cause of disruption and dysfunction of neurons, mitochondria, and endoplasmic reticulum [129,130]. Moreover, enhanced DNA fragmentation is a significant impact of acute aflatoxicosis reported in mouse testicles given a daily dosage of 2000 mg of AFB_1_ for three weeks [131]. However, a recent report investigating acute aflatoxicosis of AFB_1_ in chickens indicated that AF-dihydrodiol is an essential compound involved in acute aflatoxicosis as it produces AF-albumin adducts [132]. Furthermore, it is proposed that AFB_2_a and AFB_1_-phase-I metabolites contribute to acute toxicity. Other than that, AFB_2_a has been shown to interact with cell phospholipids and proteins, producing lipid and protein adducts and acute aflatoxicosis [133]. Notably, chronic exposure of AFs may cause similar impacts to acute toxicity; however, such impacts could be mitigated by detoxifying phase-2 enzymes and antioxidative protection pathways and DNA repairment to avoid gene mutations. On the contrary, these toxins progressively accumulate with continual exposure to small doses and develop into liver cancer such as a typical chronic exposure effect. Thus, acute aflatoxicosis can occur with a sharp accentuation of most of the harms listed above in a short period if the dosage was massive [134,135].

## 5. Strategies for Aflatoxin Mitigation

In Central America, Asia, and Africa, populations consume a high concentration of AF in their diets, as corn is a staple crop in these regions. Therefore, eradicating or reducing AF contamination is critical from a public health and economic perspective. Due to advancements in AF research over the past few decades, several novel techniques have been developed to mitigate human exposure to AFs in populations with an increased threat of aflatoxicosis; some of these strategies have been tried in real-life situations in developed nations. The primary approach is to employ genetically modified (GM) Bt corn to inhibit AF infection. The second strategy involves using NovaSil clay as a dietary supplement to aid in the absorption of AFs within the digestive system, limiting its solubility. Thirdly, non-aflatoxigenic strains of *A. flavus* (AF^−^) are used to competitively eradicate the harmful aflatoxigenic strains of *A. flavus* (AF^+^) in the field. Finally, although still in its infancy, the last method explores the usage of plant-based volatile compounds to prevent AF corruption in seeds during storage.

### 5.1. Bt Corn

Bt corn is a GM crop containing genes from *Bacillus thuringiensis* (soil-borne bacteria), encoding for Cry; the Cyt protein endotoxin, exhibiting many insecticide activities towards lepidopteran coleopteran pests, which often inhabit cereal crops [136]. Several Bt crops have been developed by inserting the *B. thuringiensis* gene, encoding endotoxins with an expanded pesticide range, and improved manifestations [137,138]. Other than corn, other GM crops include wheat, cotton, rice, peanuts, tomato, tobacco, and walnuts. Notably, immunological and metabolomics studies revealed that inclusion of the Bt protein, endotoxin Cry1Ab to swine and rats had a minimal allergic effect and slightly changed metabolomic markers and IL-6, IL-4, and CD (+) t cells production [139]. Pest invasion is closely associated with spore transmission of fungi and plant injury, resulting in a rise in fungal colonization and mycotoxin buildup.

Nevertheless, it is believed that fungal mycotoxins, such as AF, are critical for fungal protection against fungivores and other pathogens [140]. Nevertheless, another potential consequence of mycotoxins’ existence in commercial crops is that they could act as pesticides for the host plants. Nonetheless, lessening insects’ infiltration by the use of GM Bt corn provides a possible approach for reducing humans’ exposure to food-related mycotoxins. While field research in South Africa revealed a ten times reduction in fumonisin B_1_ (FB_1_) levels in Bt corn relative to non-Bt corn, the efficacy tests for Bt corn to a lower AF level in field crops exhibited varying outcomes [141]. This discrepancy could be elucidated by the presence of Bt-resistant pests on corn. Henceforth, the trials that produced adverse outcomes indicate that further study is needed to comprehend the complex mechanism of preventing AF contamination in field crops [141]. Surprisingly, the Bt Cry1Ab protein manifestation within the corn genotype did not increase AF resilience among developed testcrosses [142]. Nevertheless, the testing of attained hybrids against diverse insect stresses is yet to be conducted.

### 5.2. Biocontrol

The use of competitive non-aflatoxigenic strains of *A. flavus* and *A. parasiticus* has shown tremendous success in the biocontrol of AF contamination in both pre-and post-harvest crops [143]. Numerous field studies, especially corn, peanuts and cotton, have consistently shown a substantial reduction in AF production ranging from 70 to 90% when non-aflatoxigenic strains of *A. flavus* were used [144,145,146]. Recently, the US Environmental Protection Agency (EPA) has registered two products derived from non-aflatoxigenic strains as bio-pesticides to reduce AF contamination in cotton and peanut crops in different states of America [144]. This approach employs non-aflatoxigenic strains to competitively exclude aflatoxigenic strains that compete for agricultural resources in the same niche [147]. Cotty [148] examined the potential of AF36 (non-aflatoxigenic strains of *A. flavus*) for reducing AF contamination in cottonseed and corn, and found that it substantially decreased AF levels when co-inoculated with aflatoxigenic strains. Another non-aflatoxigenic strain, *A*. *flavus* NRRL21882 (known as Afla-Guard), was tested for AF mitigation that was highly efficient in reducing AF production in both pre-and post-harvest stages. Likewise, some other nontoxigenic *A. flavus* strains (CT3 and K49) were assessed in the US and were effective in reducing AF levels in corn [149]. In Africa, a non-aflatoxigenic strain (BN30) significantly reduced AF contamination in corn when co-inoculated with an aflatoxigenic S-strain [150,151]. In Australia, the use of non-aflatoxigenic strains of *A. flavus* markedly reduced (approximately 95%) AF contamination in peanuts [152]. In China, a non-aflatoxigenic strain of *A*. *flavus* (AF051) has lowered the population of aflatoxigenic *A. flavus* by up to 99% in peanut fields [153].

### 5.3. Clay

Clinical trials have been performed in Africa to assess the effectiveness and safety of the food supplement called NovaSil [154]. NovaSil clay comprises an extremely disinfected clay that serves as a mycotoxin absorbent in the digestive tract. NovaSil clay considerably decreased the AFB_1_-albumin adducts’ amount in blood when orally taken every day before each meal for three months. After three months of its application, the amounts of AFM_1_ were reduced by up to 60% in urine. These findings proposed that NovaSil could be used to inhibit AF-related harmful effects during prolonged dietary exposure to clay, but not for a short interval of time (one month). Other toxic compounds, such as polycyclic aromatic hydrocarbons, were also absorbed by the clay with no side effects on the liver and kidney functions [154].

### 5.4. Plants Volatiles

Despite the positive results obtained with the techniques mentioned earlier, several barriers in adopting these novel approaches for AF reduction in commercial crop production remain in developing nations of the world. For instance, there is some reluctance in growing GM crops. Additionally, industrial biohazards are related to handling vast quantities of fungal spores and the unviability of clay application as an everyday food supplement. Consequently, innovative, safe, and feasible approaches must supplement the established methods that present effectiveness and acceptability. Previous studies have proven that plant-based volatile compounds such as CO_2_ [155], ethylene [156,157], crotyl alcohol [158,159], cotton-leaf volatiles [160], and corn-based volatiles [161,162] may hinder AF’s biosynthesis. Plant-based volatile compounds in controlling AF contamination in crops are highly beneficial from a food safety perspective.

## 6. Conclusions

We have made significant progress in understanding the mechanism of AF biosynthesis, control, and adverse effects of AFs on human health. The main obstacle of existing and future research will be identifying diverse regulatory system members that relate AF biosynthesis to the unrest of cell metabolism, particularly oxidative stress. Additionally, missing enzymes in AF biosynthesis must be identified to put the puzzles of AF biosynthesis together. The analysis of the AF biosynthetic pathway resulted in the development of a safety system that defends humans from AF’s detrimental effects. Conversely, in developing countries where foodborne AFs are a source of dietary exposure, there is a need for safe, economical, and practical methods to minimize AF contamination in food.

## Figures and Tables

**Figure 1 jof-07-00606-f001:**
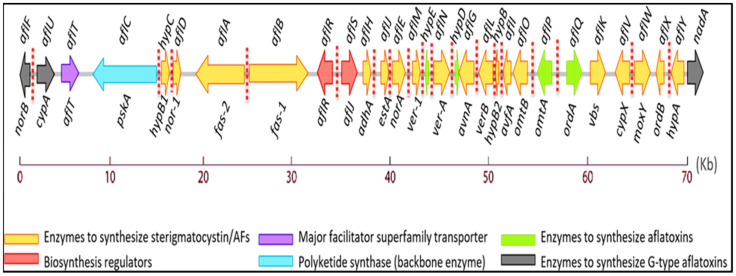
Organization of gene clusters of AFs’ biosynthesis pathway [21].

**Figure 2 jof-07-00606-f002:**
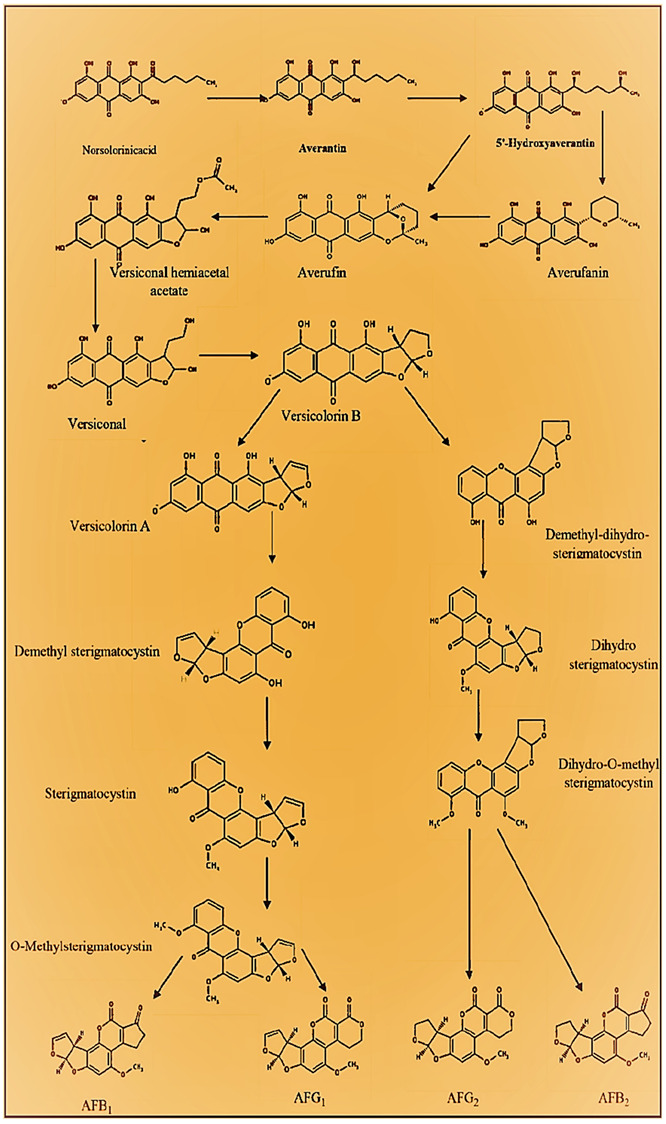
Biosynthetic pathway of AFs [22,23].

**Figure 3 jof-07-00606-f003:**
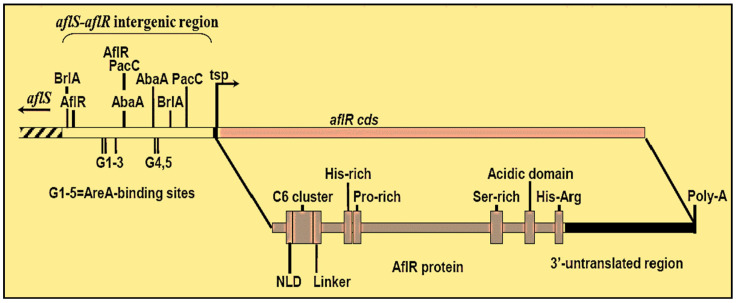
Genetic regulation of AF biosynthesis.

**Figure 4 jof-07-00606-f004:**
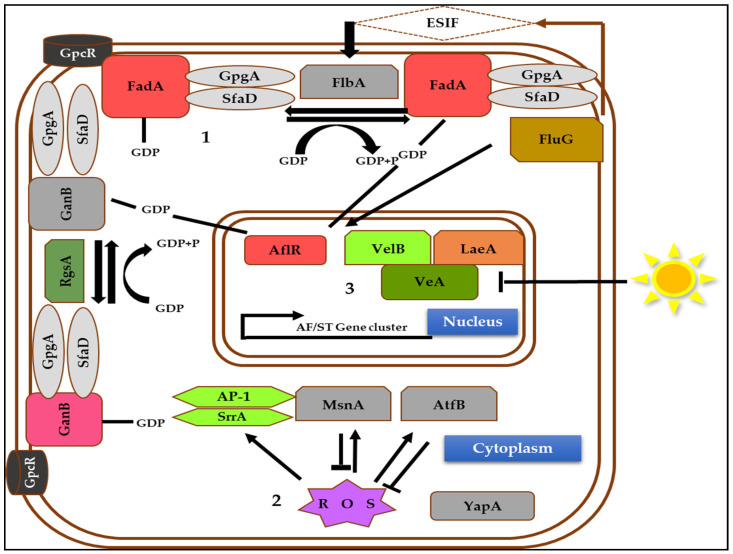
Different upstream elements affecting the AF/ST gene cluster (source: the author).

**Figure 5 jof-07-00606-f005:**
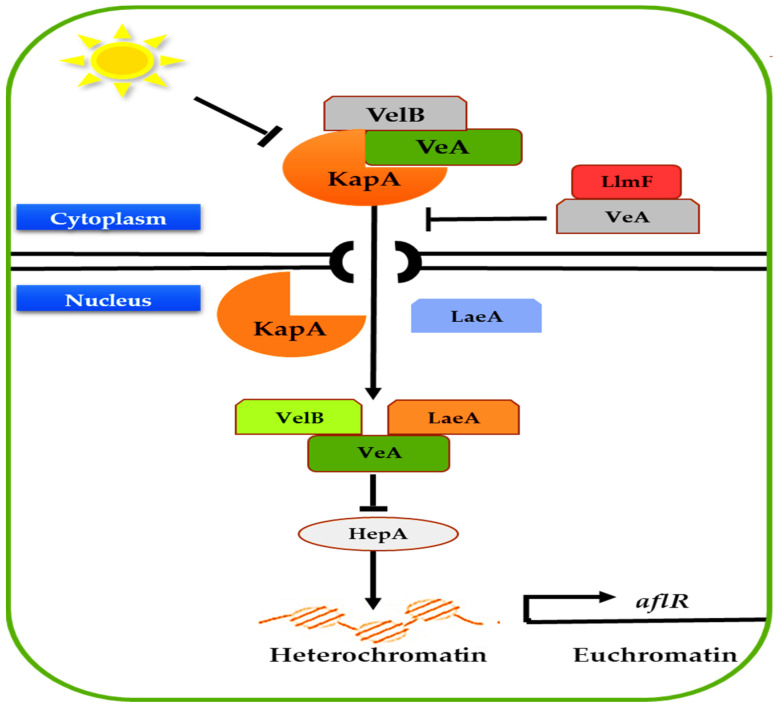
The velvet complex model (source: the author).

**Figure 6 jof-07-00606-f006:**
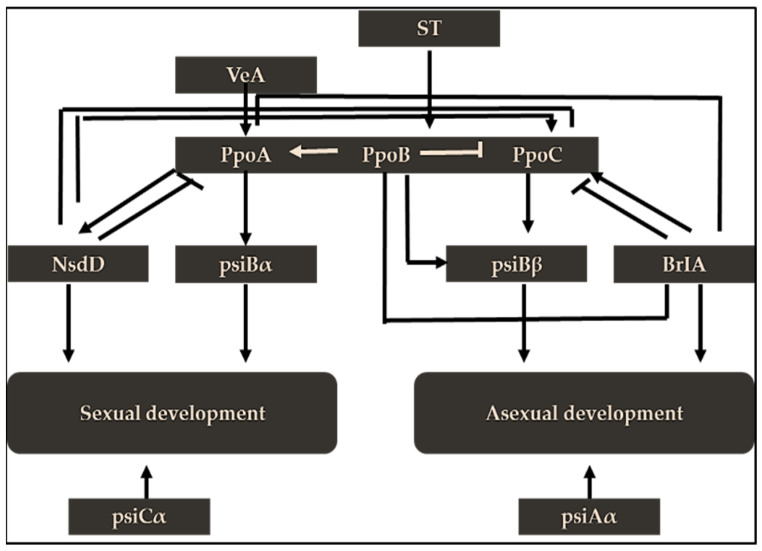
Ppo effects on ST production (source: the author).

**Figure 7 jof-07-00606-f007:**
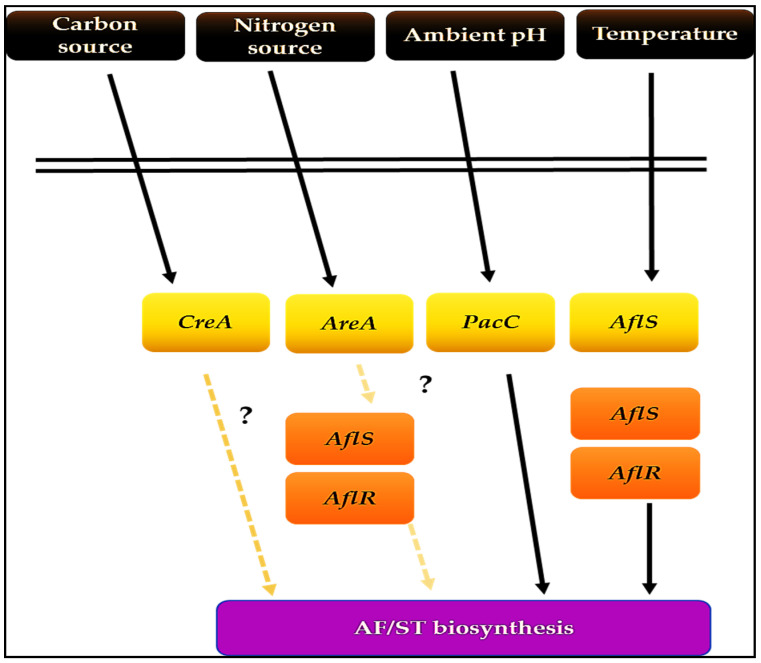
Ecological factors affecting AF/ST production. Dots signify links that have yet to be proven (source: the author).

## Data Availability

Not applicable.

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
