# Peer review of "Aflatoxin Biosynthesis, Genetic Regulation, Toxicity, and Control Strategies: A Review"

_jof, 2021, doi:10.3390/jof7080606_

Round 1
Reviewer 1 Report
The authors appear to have addressed the concerns of the reviewers in their attachment. The references have been substantially altered, with more recent references substituted for the original ones. I will presume that the authors have carefully checked that the references are properly coordinated with the text. The issue with the figures appears to have been addressed by re-drawing them. If the editor is satisfied that these new figures do not violate copyright of the original publication, the article is acceptable.
Page 6, line 173: This is “iv” (4), not “vi” (6)
Author Response
First Review Report Form
Comments and Suggestions for Authors
The authors appear to have addressed the concerns of the reviewers in their attachment. The references have been substantially altered, with more recent references substituted for the original ones. I will presume that the authors have carefully checked that the references are properly coordinated with the text. The issue with the figures appears to have been addressed by re-drawing them. If the editor is satisfied that these new figures do not violate the copyright of the original publication, the article is acceptable.
Page 6, line 173: This is “iv” (4), not “vi” (6)
Ans: Line 173: the number has been changed from vi to iv.
Reviewer 2 Report
The authors present the manuscript with the title Aflatoxin Biosynthesis, Genetic Regulation, Toxicity, and Con-2 trol Strategies: A review. The manuscript is a review focused on aflatoxins, an important group of mycotoxins. In my opinion, the manuscript is well structured, the focus on genetic regulation is a hot topic and I only have two comments:
- I recommend to rename the chapter 5 in Strategies for Aflatoxin Mitigation.
- The part 5.2. Biocontrol is not well enough presented. Being an exhaustive review, in this part I recommend to introduce more information, according to the literature published in the past years. This will make the article more citable.
Author Response
Second Review Report Form
Comments and Suggestions for Authors
The authors present the manuscript with the title Aflatoxin Biosynthesis, Genetic Regulation, Toxicity, and Control Strategies: A review. The manuscript is a review focused on aflatoxins, an important group of mycotoxins. In my opinion, the manuscript is well structured, the focus on genetic regulation is a hot topic and I only have two comments:
- I recommend renaming chapter 5 in Strategies for Aflatoxin Mitigation.
Ans: Chapter 5 is renamed as Strategies for Aflatoxin Mitigation.
- Part 5.2. Biocontrol is not well enough presented. Being an exhaustive review, in this part, I recommend introducing more information, according to the literature published in the past years. This will make the article more citable.
Ans: A new paragraph about biocontrol agents (Non-aflatoxigenic A. flavus) has been added, which is also highlighted (Lines: 443-463).
This manuscript is a resubmission of an earlier submission. The following is a list of the peer review reports and author responses from that submission.
Round 1
Reviewer 1 Report
This manuscript is a review article covering several aspects of the mycotoxin aflatoxin, including biosynthesis, regulation, toxicology and control strategies. As a review, there is no experimental protocol per se to be considered nor the validity of data analysis. Even with the limitation of the literature to only one fungal toxin, the review is extensive, quoting some 151 references. The references cover early discoveries, but for the most part, the references are from the last ten years.
The manuscript is well written, with few, if any errors in spelling or grammar. It is also well organized and concise, covering the 151 references in just 13 pages including very useful figures. Each topic is treated in order, including explanation of each step of the biosynthesis illustrated in Figure 2 and sorting the toxicity studies into types for easy comprehension since the effects of aflatoxin appear to be many.
The introduction is brief, setting the stage for the rest of the review by covering the history of the recognition of aflatoxin and the discovery of biosynthetic genes. The conclusions are also brief but suggest areas for future research.
If there are any criticisms it is that, in a few places, the statement doesn’t appear to agree with the reference number. Examples are:
Page 10, lines 316-317: Statement refers to “recent studies” but reference 108 is from 1999, which really isn’t recent.
Page 11, lines 362-364: Statement appears to refer to references 129 and 130, which are not quoted but in line 374, reference 129 (which is a clinical study of Sudanese children) modifies a statement that AF’s impair mitochondrial activity of neurons, which looks like reference 131 from the title.
Page 12, line 433: Is this really reference 142, when referring to effect of Bt corn on aflatoxin?
While many of the references are accurately placed, can the authors please check that the reference numbers are properly coordinated with the text.
Reviewer 2 Report
This ms reviews aflatoxin biosynthesis, toxicity, and control strategies, important fields of study that could help mitigate aflatoxin contamination in the food supply. The manuscript is well-written, however, I have some concerns. The ms attempts to cover several huge topics (that each would be worthy of their own review) and appears to lift figures from referenced reviews and papers with no alteration. I would suggest adapting figures, using color to indicate and differentiate processes mentioned in the text and obtaining permission for reproductions, as outlined below in the author guidelines. I would also suggest narrowing the review to the most recent advances in one of the three topics (biosynthesis, toxicity, or control strategies), as each field has made much progress in the last few years. Statements are made without references that require references. I’ve noted some below.
Figures are copied, not “adapted” e.g. Figures 4, 5, 7 are copied directly from ref 46, see attachment comparing Fig 4 (right) to the ref fig (left).
From the author guidelines:
“Your manuscript should not contain any information that has already been published. If you include already published figures or images, please obtain the necessary permission from the copyright holder to publish under the CC-BY license. For further information, see the Rights and Permissions page.
Reproducing Published Material from other Publishers
It is absolutely essential that authors obtain permission to reproduce any published material (figures, schemes, tables or any extract of a text) which does not fall into the public domain, or for which they do not hold the copyright. Permission should be requested by the authors from the copyright holder (usually the Publisher, please refer to the imprint of the individual publications to identify the copyright holder).
Permission is required for:
Use of Tables, Graphs, Charts, Schemes and Artworks if they are unaltered or slightly modified.”
Need references for lines: 102, 138, 440-446, 447-457
